# OpenReview forum: "META-LEARNING FOR SCIENTIFIC HYPOTHESIS GENERATION AND EXPERIMENTAL DESIGN"
_ICLR.cc/2025/Workshop/AgenticAI — ICLR 2025 Workshop AgenticAI Reject_

### Official Review · Reviewer_4fip · 2025-02-28
**Need more work**

**Rating:** 3
**Confidence:** 4

**Review:**

This paper proposes a meta-learning framework combining few-shot learning and reinforcement learning for scientific hypothesis generation and experimental design across multiple domains.

Strengths:
1. Addresses an important problem with potential cross-domain applications

Weaknesses:
1. Lacks Technical Details:
No concrete algorithms or implementation details.
Results appear aspirational rather than from completed experiments.
Claims statistical significance without methodological details.
2. Insufficient Evaluation:
Performance claims lack proper experimental validation.
Baseline comparisons are superficial.
No specific details on how experiments were conducted.

Overall, I feel this paper needs to be significantly polished and expanded for details.

---

### Official Review · Reviewer_t4k1 · 2025-02-28
**Not good enough.**

**Rating:** 3
**Confidence:** 4

**Review:**

This paper works on a meta-learning framework to accelerate hypothesis generation and experimental design using agentic AI systems. The proposed approach can enable AI agents to learn across diverse scientific domains with minimal labeled data adapting different domains. Overall, the problem is innovative but the paper is not in good quality and the experiments and the analysis are not sufficient to prove the effectiveness of the proposed method.

quality: The paper is not fully developed with no clear clarification about the prior work and detailed explanation of the methodology. The experiments cover different scientific domains which are good but the results show only one domain with not sufficient analysis.

clarity: The paper clearly shows the challenges of using AI for scientific discoveries. However, it does not clearly describe the proposed framework and the experimental setup and evaluation process. Moreover, the experiments are not completely finished and are not sufficient to show the effectiveness of the proposed framework.

originality: The problem and methodologies proposed in this paper are novel and important for AI in scientific discoveries.

Pros:
1. The problem of using AI agents to enhance scientific discovery is important and the methodologies shown in this paper are novel.
2. The tasks and datasets in this paper show a diversity of scientific discovery domains.

Cons:
1. The methodology behind this paper is not clearly presented.
2. The experimental results are not sufficient enough to show the effectiveness of the proposed framework.
3. The paper is not well-written and can be further improved.

---

### Official Review · Reviewer_6M7T · 2025-03-02
**Serious Concerns on Structure, Rigor, and Authenticity**

**Rating:** 2
**Confidence:** 5

**Review:**

Summary:

This paper presents a meta-learning framework that integrates few-shot learning and reinforcement learning (RL) to automate scientific hypothesis generation and experimental design. The authors claim that their approach accelerates discovery across multiple domains, including materials science, drug discovery, and physics simulations. However, the manuscript primarily consists of bullet points rather than fully developed text, raising serious concerns regarding its originality, coherence, and scientific rigor.

Weaknesses:

1. The manuscript is largely composed of bullet points, and lacks detailed explanations, making it difficult to assess its depth and clarity.

2. It fails to provide essential sections typically expected in a scientific paper, such as a well-structured Introduction, a clear Problem Statement, a thorough Related Work review, detailed Implementation and Experimental Methods, and a meaningful Result Analysis. The absence of these critical components gives the impression of minimal human effort or intellectual contribution.

3. A manual verification of citations revealed discrepancies. For instance, the second reference in the bibliography— Tabor, Z., Newton, K. F., & Brightman, I. (2022). Automated hypothesis generation in materials design using few-shot neural architectures. Advanced Materials, 34(12), 2106813.—could not be found despite extensive searching, raising concerns about citation integrity.

Overall, this paper exhibits signs of potential AI-generated content. Even if human-authored, it suffers from fundamental weaknesses in theoretical grounding, empirical validation, and scientific contribution, rendering it unsuitable for publication.

---

### Decision · Program_Chairs · 2025-03-05

Reject